# Nasopharyngeal carriage of *Streptococcus pneumoniae* among children <5 years of age in Indonesia prior to pneumococcal conjugate vaccine introduction

Dodi Safari[1]*, Wa Ode Dwi Daningrat[1,2], Jennifer L. Milucky[3], Miftahuddin Majid Khoeri[1], Wisiva Tofriska Paramaiswari[1], Wisnu Tafroji[1], Korrie Salsabila[1], Yayah Winarti[1], Amin Soebandrio[1], Sri Rezeki Hadinegoro[4], Ari Prayitno[4], Lana Childs[5], Fabiana C. Pimenta[3], Maria da Gloria Carvalho[3], Tamara Pilishvili[3¤]

1 Eijkman Research Center for Molecular Biology, National Research and Innovation Agency, Cibinong, West Java, Indonesia, 2 Centre for Genomic Pathogen Surveillance, Big Data Institute, Nuffield Department of Medicine, University of Oxford, Oxford, United Kingdom, 3 National Center for Immunization and Respiratory Diseases, Centers for Disease Control and Prevention, Atlanta, GA, United States of America, 4 Faculty of Medicine, University of Indonesia, Jakarta, Indonesia, 5 CDC Foundation, Atlanta, GA, United States of America

¤ Current address: Medical and Clinical Affairs, US GSK Vaccines, GSK, Philadelphia, PA, United States of America
* dodi004@brin.go.id

**Data Availability Statement:** All relevant data are within the manuscript and its Supporting Information files.

## Abstract

Pneumococcal conjugate vaccines (PCVs) prevent nasopharyngeal colonization with vaccine serotypes of *Streptococcus pneumoniae*, leading to reduced transmission of pneumococci and stronger population-level impact of PCVs. In 2017 we conducted a cross-sectional pneumococcal carriage study in Indonesia among children aged <5 years before 13-valent PCV (PCV13) introduction. Nasopharyngeal swabs were collected during visits to community integrated health service posts at one peri-urban and one rural study site. Specimens were analyzed by culture, and isolates were serotyped using sequential multiplex polymerase chain and Quellung reaction. Antibiotic susceptibility was performed by broth microdilution method. We enrolled 1,007 children in Gunungkidul District, Yogyakarta (peri-urban) and 815 in Southwest Sumba, East Nusa Tenggara (rural). Pneumococcal carriage prevalence was 30.9% in Gunungkidul and 87.6% in Southwest Sumba (combined: 56.3%). PCV13 serotypes (VT) carriage was 15.0% in Gunungkidul and 52.6% in Southwest Sumba (combined: 31.8%). Among pneumococcal isolates identified, the most common VT were 6B (16.4%), 19F (15.8%), and 3 (4.6%) in Gunungkidul (N = 323) and 6B (17.6%), 19F (11.0%), and 23F (9.3%) in Southwest Sumba (N = 784). Factors associated with pneumococcal carriage were age (1–2 years adjusted odds ratio (aOR) 1.9, 95% CI 1.4–2.5; 3–4 years aOR 1.5, 95% CI 1.1–2.1; reference <1 year), other children <5 years old in the household (aOR 1.5, 95% CI 1.1–2.0), and presence of ≥1 respiratory illness symptom (aOR 1.8, 95% CI 1.4–2.2). Overall, 61.5% of the pneumococcal isolates were non-susceptible to ≥1 antibiotic class and 13.2% were multi-drug non-susceptible (MDNS) (non-susceptible to ≥3 classes of antibiotics). Among 602 VT isolates, 73.9% were non-susceptible

**Funding:** This work was funded by the Grant or Cooperative Agreement Number, NU2GGH001852-03, funded by the Centers for Disease Control and Prevention. Its contents are solely the responsibility of the authors and do not necessarily represent the official views of the Centers for Disease Control and Prevention or the Department of Health and Human Services. The funders had no role in the study design, data collection and analysis, decision to publish, or preparation of the manuscript.

**Competing interests:** The authors have declared that no competing interests exist.

and 19.9% were MDNS. These findings are critical to establish a pre-PCV13 carriage prevalence and demonstrate the complexity in evaluating the impact of PCV13 introduction in Indonesia given the wide variability in the carriage prevalence as shown by the two study sites.

## Introduction

*Streptococcus pneumoniae* (pneumococcus) is a leading cause of bacterial pneumonia, meningitis, and sepsis among children worldwide [1]. In 2015, an estimated 9.2 million cases of pneumococcal infections and 318,000 associated deaths occurred in children <5 years of age worldwide; of these 4.4 million pneumococcal cases and 88,500 deaths occurred in Southeast Asia [2]. Prior to introduction of pneumococcal conjugate vaccines (PCVs), children <2 years of age were at the highest risk of serious pneumococcal infections, and six to 11 serotypes were responsible for ≥70% of invasive pneumococcal disease (IPD) burden globally [3]. Introduction of PCVs in many countries led to significant reductions in IPD among vaccinated children, as well as unvaccinated older populations through indirect (or herd) effects [4–6].

To date, 165 countries have introduced PCVs into their routine infant immunization schedule [7]. Prior to 2017, 10-valent PCV (PCV10) and the 13-valent PCV (PCV13) were available in Indonesia as part of a private service in hospitals. In 2017, the government of Indonesia introduced PCV13 in a limited geographic area, Lombok Island in the West Nusa Tenggara province, as a demonstration project using a schedule of two primary doses at two and three months of age followed by a booster at 12 months (2+1 schedule) [8]. In 2021, PCV13 was introduced as a part of national program and launched in select districts in East and West Java with nationwide introduction starting in 2022.

Pneumococcal colonization is a precursor of infection, and PCVs reduce vaccine serotype pneumococcal colonization among vaccinated children, which leads to decreased transmission of vaccine serotypes in communities [9]. Nasopharyngeal colonization studies can provide information on pneumococcal serotypes circulating in the community and help document the impact of PCV immunization programs among vaccinated children (direct effects) and unvaccinated children through reduced transmission of vaccine serotypes from vaccinated population groups (indirect effects) [10, 11]. In 2017, prior to PCV introduction, we conducted a pneumococcal carriage study to evaluate nasopharyngeal colonization with *S. pneumoniae* among children <5 years of age in two distinct communities in Indonesia. We evaluated overall rates of pneumococcal carriage, factors associated with pneumococcal carriage, serotype distribution, and antibiotic susceptibility of *S. pneumoniae* strains. The findings of this evaluation will provide baseline data to evaluate the potential impact of PCVs in Indonesia.

## Methods

### Study design and population

We conducted a cross-sectional survey of nasopharyngeal pneumococcal colonization among children <5 years of age between February–May 2017 in two distinct geographic regions of Indonesia: Southwest Sumba in East Nusa Tenggara, a rural area in the East region, and Gunungkidul in Yogyakarta, a peri-urban area in the West region. Gunungkidul district, Yogyakarta province, has an estimated total population of 770,880 in 2022, with 46,958 (6%) children <5 years of age. Southwest Sumba district in the province of East Nusa Tenggara has an estimated total population of 308,106 in 2022, with 41,334 (13%) children <5 years of age. The district of Southwest Sumba is primarily comprised of rural sub-districts with less access to

health services, clean water, and education. Although parts of Gunungkidul district were also categorized as rural by Statistics Indonesia, generally, the district has better access to health services, clean water, and education compared to Southwest Sumba [12, 13].

Study documents were reviewed and approved by Eijkman Institute Research Ethics Commission (Protocol No. 104) and the U.S. Centers for Disease Control and Prevention (CDC) Institutional Review Board (Protocol 6940). Written informed consent from the parent/guardian of each enrolled child was obtained.

## Data and sample collection

We enrolled children <5 years of age presenting for preventive care at community integrated health service posts (posyandu). Children were not eligible to participate if the parent/guardian identified the child as unwell or the child was in obvious distress, the parent/guardian was not conversant in Bahasa, the official language of Indonesia, or the parent/guardian did not provide informed consent. Trained staff interviewed parents/guardians to collect information on demographics, household exposures to smoke, recent illness episodes, and recent antibiotic use. Vaccination history was obtained from official vaccination cards (mother-child health book) or records kept at the posyandu.

Nasopharyngeal specimens were collected by trained medical staff using nylon flocked swabs [Ultra minitip pediatric (COPAN; Cat. No. 516CS01)] for children <1 years old and flexible minitip (COPAN; Cat. No. 503CS01) for children ≥1 years old. Swabs were placed into 1.0 ml skim milk tryptone glucose glycerol (STGG) medium and stored at -80°C. Specimens were tested at the Eijkman Institute for Molecular Biology in Jakarta.

## *Streptococcus pneumoniae* identification, serotyping, and antimicrobial susceptibility testing

For NP swab analysis, 200μl of swab-inoculated STGG media was transferred to 5.0 ml Todd Hewitt broth containing 0.5% yeast extract (THY), and 1ml of rabbit serum and incubated at 35–37°C for six hours. Cultured broth was plated on sheep blood agar and incubated in 5% $CO_2$ at 35–37°C. After 18–24 hours of incubation, plates were examined for the appearance of alpha-hemolytic colonies resembling streptococci. Pneumococci were identified by susceptibility to optochin and bile solubility. *S. pneumoniae* isolates were serotyped by conventional multiplex polymerase chain reaction (cmPCR) based testing [14–17] followed by Quellung reaction (Staten Institute, Denmark). Non-typeable isolates were confirmed as *S. pneumoniae* by real-time PCR, targeting *lyt*A gene [18].

Antimicrobial susceptibility to moxifloxacin, penicillin, levofloxacin, meropenem, azithromycin, tetracycline, ertapenem, erythromycin, cefuroxime, amoxicillin/clavulanic Acid 2:1, trimethoprim-sulfamethoxazole (SXT), ceftriaxone, linezolid, vancomycin, cefotaxime, clindamycin, cefepime, and chloramphenicol was determined by broth microdilution method using commercial minimal inhibitory concentration (MIC) plate (STP6F Trek Diagnostics, Thermo Fisher Scientific, USA) according to manufacturer's instructions. Isolates were classified as susceptible, intermediate, or resistant according to the 2022 Clinical and Laboratory Standards Institute (CLSI) guidelines [19]. For the penicillin interpretive categories, we used the parenteral non-meningitis and meningitis MIC breakpoints (S1 File). Similarly for ceftriaxone and cefepime, we used the non-meningitis and meningitis MIC breakpoints (S1 File). Isolates intermediate or resistant to ≥1 antibiotic were classified as non-susceptible (NS). Multi-drug non-susceptibility (MDNS) was defined as NS (i.e., intermediate or resistant) to ≥3 classes of antibiotics [19]. When classifying isolates as NS or MDNS, we used the non-meningitis breakpoints for penicillin, ceftriaxone, and cefepime.

## Data analysis

The sample size was estimated to allow for 90% power to detect a statistically significant ($\alpha$ = 0.05) 50% decline in the proportion of children colonized with vaccine serotypes in a post-PCV13 introduction carriage survey among children aged <5 years old, assuming vaccine coverage of at least 80%. A nasopharyngeal pneumococcal carriage study done in Lombok in 2012 found that approximately 46% of children aged <5 years were colonized with *S. pneumoniae* with approximately 50% of those carrying PCV13 serotypes. Under the above assumptions, we estimated the required sample size to be 324 children <1 year of age and 986 children 1–4 years of age. The required sample size was adjusted during the first few weeks of data collection based on the differences in the preliminary overall pneumococcal carriage prevalence observed in each study site.

We calculated pneumococcal carriage prevalence among children by study site (Gunungkidul and Southwest Sumba), age group (<1 year, 1–2 years, 3–4 years), and other factors potentially associated with pneumococcal colonization. The vaccine serotypes were those included in PCV13 (1, 3, 4, 5, 6A, 6B, 7F, 9V, 14, 18C, 19A, 19F, and 23F). *S. pneumoniae* isolates were deemed to be non-typeable if a serotype could not be determined by cmPCR and Quellung but showed positive results for *lytA* gene.

We used logistic regression models to obtain crude odds ratios (ORs) and adjusted ORs (adjusted by study site) to evaluate factors associated with pneumococcal carriage prevalence. We applied a multivariable logistic regression model to evaluate factors independently associated with pneumococcal carriage; a stepwise backwards elimination process was applied to the full model including all covariates, and variables were removed from the model when 0.05 significance level was not met. Data analyses were performed using Stata (version 15.0) and SAS (version 9.4) software. *P* values <0.05 were considered statistically significant.

## Results

### Participant characteristics

Between February–May 2017, we enrolled 1,822 children <5 years of age (Gunungkidul: 1,007; Southwest Sumba: 815); of these, 449 (24.6%) were <1 year of age, 762 (41.8%) were 1–2 years of age, and 611 (33.5%) were 3–4 years of age. Characteristics of enrolled children by study site are presented in Table 1. A higher proportion of children in Southwest Sumba as compared to those from Gunungkidul lived in households with >6 members per household (44.8% vs. 12.9%, *P*<0.0001) and with other children <5 years of age (31.8% vs. 13.6%, *P*<0.0001), had reported having cough (53.3% vs. 14.5%, *P*<0.0001), runny nose (67.7% vs. 21.9%, *P*<0.0001), difficulty breathing (12.8% vs. 0.1%, *P*<0.0001), or fever (25.8% vs. 5.8%, *P*<0.0001) in the last 24 hours, and had taken antibiotics during the past three days (7.6% vs. 5.1%, *P* = 0.02). We found that a higher proportion of children attended daycare in Gunungkidul (30.7%) compared to Southwest Sumba (17.8%, *P*<0.0001). The primary fuel source used for cooking in Gunungkidul was liquefied petroleum gas (LPG) or kerosene (62.5%) whereas in Southwest Sumba almost all households used wood only (96.1%, *P*<0.0001). In both study sites, nearly all households cooked indoors (Gunungkidul: 97.3%; Southwest Sumba: 98.2%).

### *S. pneumoniae* carriage and factors associated with pneumococcal carriage rates

The overall prevalence of *S. pneumoniae* carriage was 56.3% (1,025/1,822). We found the carriage prevalence was 87.6% and 30.9% among children in Southwest Sumba (714/815) and Gunungkidul (311/1,007) (*P*<0.0001), respectively (Table 1). Prevalence of *S. pneumoniae*

**Table 1. Participant characteristics and *S. pneumoniae* carriage prevalence overall and by study site.**

| Characteristic | Gunungkidul | | Southwest Sumba | | Total | | P value (participant characteristic by study site) |
|---|---|---|---|---|---|---|---|
| | Tested n (%) | *S. pneumoniae* detected n (%) | Tested n (%) | *S. pneumoniae* detected n (%) | Tested n (%) | *S. pneumoniae* detected n (%) | |
| **Study site** | 1007 | 311 (30.9) | 815 | 714 (87.6) | 1822 | 1025 (56.3) | |
| **Sex** | | | | | | | |
| Male | 515 (51.1) | 166 (32.2) | 417 (51.2) | 361 (86.6) | 932 (51.2) | 527 (56.5) | 0.99 |
| Female | 492 (48.9) | 145 (29.5) | 398 (48.8) | 353 (88.7) | 890 (48.8) | 498 (56.0) | |
| **Age (year)** | | | | | | | |
| <1 year old | 243 (24.1) | 51 (21.0) | 206 (25.3) | 175 (85.0) | 449 (24.6) | 226 (50.3) | 0.04 |
| 1–2 years old | 401 (39.8) | 144 (35.9) | 361 (44.3) | 324 (89.8) | 762 (41.8) | 468 (61.4) | |
| 3–4 years old | 363 (36.0) | 116 (32.0) | 248 (30.4) | 215 (86.7) | 611 (33.5) | 331 (54.2) | |
| **Presence of other children <5 years old** | | | | | | | |
| Yes | 137 (13.6) | 55 (40.1) | 259 (31.8) | 231 (89.2) | 396 (21.7) | 286 (72.2) | <0.0001 |
| No | 870 (86.4) | 256 (29.4) | 556 (68.2) | 483 (86.9) | 1426 (78.3) | 739 (51.8) | |
| **Household size** | | | | | | | |
| 2–3 | 109 (10.8) | 33 (30.3) | 64 (7.9) | 59 (92.2) | 173 (9.5) | 92 (53.2) | <0.0001 |
| 4–6 | 768 (76.3) | 235 (30.6) | 386 (47.4) | 336 (87.0) | 1154 (63.3) | 571 (49.5) | |
| >6 | 130 (12.9) | 43 (33.1) | 365 (44.8) | 319 (87.4) | 495 (27.2) | 362 (73.1) | |
| **Primary fuel** | | | | | | | |
| LPG/kerosene only | 629 (62.5) | 198 (31.5) | 27 (3.3) | 23 (85.2) | 656 (36.0) | 221 (33.7) | <0.0001 |
| Wood only | 180 (17.9) | 61 (33.9) | 783 (96.1) | 687 (87.7) | 963 (52.9) | 748 (77.7) | |
| Wood with any other source | 198 (19.7) | 52 (26.3) | 5 (0.6) | 4 (80.0) | 203 (11.1) | 56 (27.6) | |
| **Cooking place** | | | | | | | |
| Inside the house | 980 (97.3) | 298 (30.4) | 800 (98.2) | 702 (87.8) | 1780 (97.7) | 1000 (56.2) | 0.23 |
| Outside the house | 27 (2.7) | 13 (48.1) | 15 (1.8) | 12 (80.0) | 42 (2.3) | 25 (59.5) | |
| **Breastfeeding status** | | | | | | | |
| Never breastfed | 25 (2.5) | 9 (36.0) | 16 (2.0) | 13 (81.3) | 41 (2.3) | 22 (53.7) | 0.10 |
| Currently breastfed | 436 (43.3) | 125 (28.7) | 317 (38.9) | 276 (87.1) | 753 (41.3) | 401 (53.3) | |
| Ever breastfed | 546 (54.2) | 177 (32.4) | 482 (59.1) | 425 (88.2) | 1028 (56.4) | 602 (58.6) | |
| **Daycare attendance** | | | | | | | |
| Yes | 309 (30.7) | 99 (32.0) | 145 (17.8) | 128 (88.3) | 454 (24.9) | 227 (50.0) | <0.0001 |
| No | 698 (69.3) | 212 (30.4) | 670 (82.2) | 586 (87.5) | 1368 (75.1) | 798 (58.3) | |
| **Exposure to cigarette smoke in the household** | | | | | | | |
| Yes | 711 (70.6) | 212 (29.8) | 558 (68.5) | 485 (86.9) | 1269 (69.6) | 697 (54.9) | 0.32 |
| No | 296 (29.4) | 99 (33.4) | 257 (31.5) | 229 (89.1) | 553 (30.4) | 328 (59.3) | |
| **Current illness (in the last 24 hours)** | | | | | | | |
| Cough | 146 (14.5) | 54 (37.0) | 434 (53.3) | 393 (90.6) | 580 (31.8) | 447 (77.1) | <0.0001 |
| Runny nose | 221 (21.9) | 95 (43.0) | 552 (67.7) | 500 (90.6) | 773 (42.4) | 595 (77.0) | <0.0001 |
| Difficulty breathing | 1 (0.1) | 0 (0.0) | 104 (12.8) | 95 (91.3) | 105 (5.8) | 95 (90.5) | <0.0001 |
| Fever (in the last 3 days) | 58 (5.8) | 19 (32.8) | 210 (25.8) | 187 (89.0) | 268 (14.7) | 206 (76.9) | <0.0001 |
| **Hospital admission in the past 3 months** | | | | | | | |
| Yes | 23 (2.3) | 9 (39.1) | 26 (3.2) | 23 (88.5) | 49 (2.7) | 32 (65.3) | 0.23 |
| No | 984 (97.7) | 302 (30.7) | 789 (96.8) | 691 (87.6) | 1773 (97.3) | 993 (56.0) | |
| **Any antibiotics used** | | | | | | | |
| During the past 3 days | 51 (5.1) | 16 (31.4) | 62 (7.6) | 57 (91.9) | 113 (6.2) | 73 (64.6) | 0.02 |
| During the past 30 days | 120 (11.9) | 34 (28.3) | 123 (15.1) | 109 (88.6) | 243 (13.3) | 143 (58.8) | 0.05 |

carriage was higher in children aged 1–2 years (61.4%; 468/762) than children <1 year (50.3%; 226/449) or 3–4 years (54.2%; 331/611) (Table 1). Pneumococcal carriage prevalence was higher in children who lived with >6 household members (73.1%; 362/495) compared to those children residing in households with 4–6 (49.5%; 571/1,154) or 2–3 (53.2%; 92/173) household members (Table 1), and in households with other children <5 years of age (72.2%; 286/396 vs. 51.8%; 739/1,426). Children who did not attend daycare also had a slightly higher carriage prevalence than those who did (58.3%; 798/1,368 vs. 50.0%; 227/454); however, when stratified by site there was no difference in the carriage prevalence by daycare attendance (Gunungkidul: 32.0%; 99/309 vs. 30.4%; 212/698; Southwest Sumba: 88.3%; 128/145 vs. 87.5%; 586/670). Carriage prevalence was higher among children from households using wood as primary fuel (77.7%; 748/963) compared to those using natural gas (33.7%; 221/656) (Table 1). Children reporting cough, runny nose, difficulty breathing, or fever in the past 24 hours had a higher carriage prevalence than those without these symptoms (Table 1).

After adjusting for study site, age (1–2 years adjusted OR = 1.9, 95% CI = 1.4–2.5; 3–4 years adjusted OR = 1.5, 95% CI = 1.1–2.1), presence of other children <5 years old (adjusted OR = 1.5, 95% CI = 1.1–2.0), and presence of ≥1 symptom of respiratory illness (adjusted OR = 1.8, 95% CI = 1.4–2.2) were significantly associated with pneumococcal carriage. In a multivariate analysis, the odds of pneumococcal carriage varied significantly by several characteristics: children aged 1–2 years had a 1.7-fold-increased odds compared to children aged <1 year ($P$ = 0.0008), households with the presence of other children <5 years old had a 1.9-fold-increased odds compared to those without ($P$<0.0001), households with 4–6 persons had a 0.8-fold-decreased odds compared to households with 2–3 persons ($P$ = 0.012), households using wood only as the primary fuel source had a 4.8-fold-increased odds ($P$<0.0001) while households using wood with any other source had a 0.8-fold-decreased odds ($P$<0.0001) compared to households using LPG or kerosene only, exposure to cigarette smoke in the household was associated with a 0.8-fold-decreased odds ($P$ = 0.034), and the presence of ≥1 symptom of respiratory illness was associated with a 2.8 fold-increased odds ($P$<0.0001) (Table 2).

## Serotype distribution

A total of 1,107 *S. pneumoniae* isolates were obtained from 1,025 children colonized with pneumococcus; 784 (70.8%) isolates were obtained from Southwest Sumba and 323 (29.2%) from Gunungkidul. We identified 81/1,025 children (7.9%) colonized with >1 *S. pneumoniae* strain (80 with two strains and one with three strains). Among these children, 35/81 (43.2%) were co-colonized with a non-typeable strain and a vaccine serotype or non-vaccine serotype, 21/81 (25.9%) with two vaccine serotypes, 18/81 (22.2%) with a vaccine and non-vaccine serotype, 6/81 (7.4%) with two non-vaccine serotypes, and 1/81 (1.2%) with two vaccine serotypes and one non-vaccine serotype. Overall, the proportion of children colonized with at least one vaccine serotype was 31.8% (54.4% of isolates, n = 602); vaccine serotype carriage was 52.6% (57.1% of isolates, n = 448) in Southwest Sumba and 15.0% (47.7% of isolates, n = 154) in Gunungkidul (S1 and S3 Figs). A higher proportion of children 1–2 years of age (37.1%, 57.9% of isolates) were colonized with a vaccine serotype than children <1 year of age (26.5%, 49.8% isolates) or children 3–4 years of age (29.1%, 52.5% of isolates) ($P$ = 0.0001) (S2 and S3 Figs). In Southwest Sumba, vaccine serotype carriage was higher in children 1–2 years of age (60.9%, 51.1% of isolates) than children <1 year (46.1%, 21.9% of isolates) and 3–4 years of age (46.0%, 27.0% of isolates) ($P$ = 0.0001) (S3 Fig). In Gunungkidul, vaccine serotype colonization rates were the highest among children 3–4 years of age (17.6%, 42.2% of isolates) compared to the other two age groups (<1 year: 9.9%, 15.6% of isolates; 1–2 years: 15.7%, 42.2% of isolates) ($P$ = 0.03) (S3 Fig). Among the 1,107 pneumococcal isolates identified, the most common

**Table 2. Factors associated with *S. pneumoniae* colonization in children <5 years of age.**

| Characteristic | Overall | | | | | |
|---|---|---|---|---|---|---|
| | Crude OR (95% CI) | p value | Adjusted by study site OR (95% CI)[a] | p value | Multivariate OR (95% CI)[b] | p value |
| **Sex** | | | | | | |
| Female | 1.0 (0.8–1.2) | 0.800 | 1.0 142.8–1.2) | 0.763 | | |
| **Age (year)** | | | | | | |
| <1 year | *Ref* | | *Ref* | | *ref* | |
| 1–2 years old | 1.6 (1.2–2.0) | 0.0002 | 1.9 (1.4–2.5) | <0.0001 | 1.7 (1.3–2.2) | 0.0008 |
| 3–4 years old | 1.2 (0.9–1.5) | 0.216 | 1.5 (15.1–2.1) | 0.006 | 1.3 (1.0–1.7) | 0.962 |
| **Presence of other children <5 years old** | | | | | | |
| Yes | 2.4 (1.9–3.1) | <0.0001 | 1.5 (1.1–2.0) | 0.012 | 1.9 (1.4–2.5) | <0.0001 |
| **Household size** | | | | | | |
| 2–3 | *Ref* | | *Ref* | | *ref* | |
| 4–6 | 0.9 (0.6–1.2) | 0.365 | 0.9 (0.6–1.3) | 0.620 | 0.8 (0.5–1.1) | 0.012 |
| >6 | 2.4 (1.7–3.4) | <0.0001 | 1.0 (0.6–1.5) | 0.864 | 1.1 (0.7–1.7) | 0.167 |
| **Primary fuel** | | | | | | |
| LPG/kerosene only | *Ref* | | *Ref* | | *ref* | |
| Wood Only | 6.8 (5.5–8.5) | <0.0001 | 1.1 (0.8–1.6) | 0.470 | 4.8 (3.8–6.2) | <0.0001 |
| Wood with any other source | 0.8 (0.5–1.1) | 0.105 | 0.8 (0.5–1.1) | 0.158 | 0.8 (0.5–1.1) | <0.0001 |
| **Cooking place** | | | | | | |
| Inside the house | 0.9 (0.5–1.6) | 0.666 | 0.6 (0.3–1.3) | 0.192 | | |
| **Breastfeeding status** | | | | | | |
| Never breastfed | *Ref* | | *Ref* | | | |
| Currently breastfed | 1.0 (0.5–1.8) | 0.960 | 0.9 (0.4–1.9) | 0.743 | | |
| Ever breastfed | 1.2 (0.7–2.3) | 0.533 | 1.0 (0.5–2.2) | 0.940 | | |
| **Daycare attendance** | | | | | | |
| Yes | 0.7 (0.6–0.9) | 0.002 | 1.1 (0.8–1.4) | 0.554 | | |
| **Exposure to cigarette smoke in the household** | | | | | | |
| Yes | 0.8 (0.7–1.0) | 0.083 | 0.8 (0.7–1.1) | 0.152 | 0.8 (0.6–1.0) | 0.034 |
| **Current illness (in the last 24 hours)** | | | | | | |
| Presence of ≥1 symptom of respiratory Illness | 4.5 (3.7–5.5) | <0.0001 | 1.8 (1.4–2.2) | <0.0001 | 2.8 (2.2–3.4) | <0.0001 |
| **Hospital admission in the past 3 months** | | | | | | |
| Yes | 1.5 (0.8–2.7) | 0.198 | 1.3 (0.6–2.7) | 0.440 | | |
| **Any antibiotics used** | | | | | | |
| During the past 3 days | 1.5 (1.0–2.2) | 0.065 | 1.2 (0.7–1.9) | 0.470 | | |
| During the past 30 days | 1.1 (0.9–1.5) | 0.383 | 0.9 (0.7–1.3) | 0.753 | | |

OR: odds ratio

[a]Odds ratio adjusted by study site.

[b]Stepwise backwards elimination process was applied to the full model including all covariates, and variables were removed from the model when 0.05 significance level was not met.

vaccine serotypes were 6B (17.3%), 19F (12.4%), and 23F (7.8%), and the most common non-vaccine serotypes were 6C (5.0%), 11A (4.0%), and 34 (3.1%). In Gunungkidul, among the 323 pneumococcal isolates identified, the most common vaccine serotypes were 6B (16.4%), 19F (15.8%), and 3 (4.6%), and the most common non-vaccine serotypes were 6C (11.1%), 34 (6.8%), and 15C (3.4%) (Fig 1). In Southwest Sumba, among the 784 pneumococcal isolates

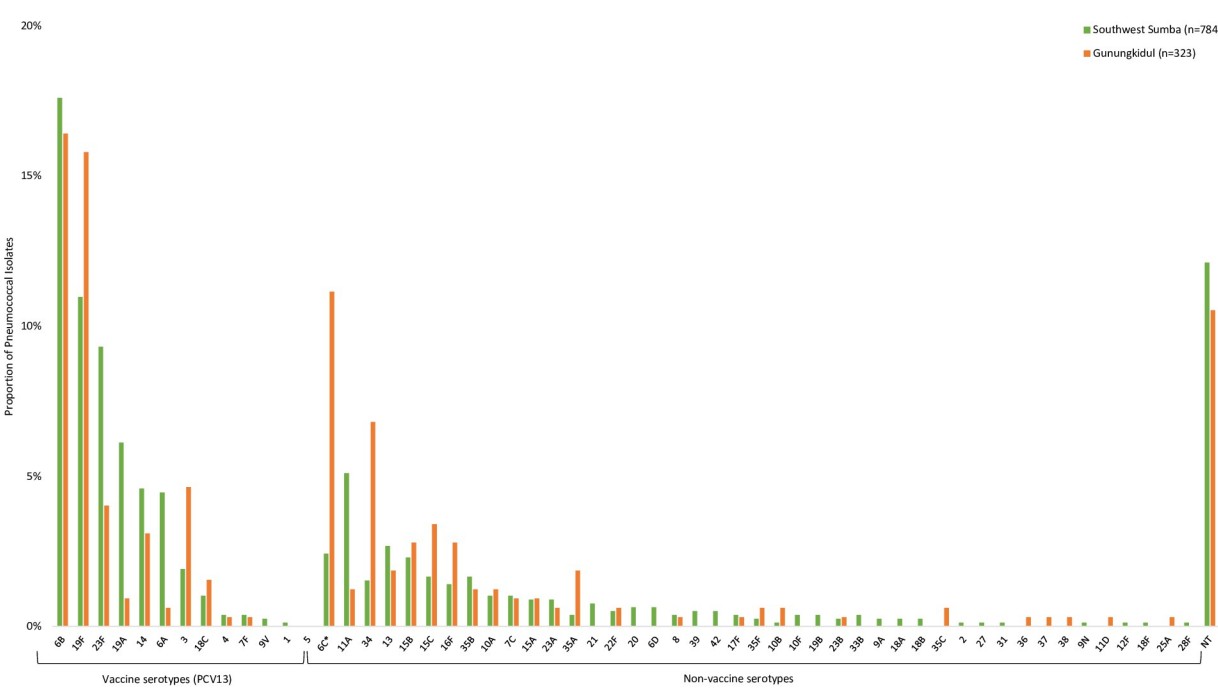

*Cross-protection is expected from 6A antigen in PCV13 [45].

**Fig 1. Serotype distribution of *Streptococcus pneumoniae* Isolates in Gunungkidul and Southwest Sumba, Indonesia.** *Cross-protection is expected from 6A antigen in PCV13 [45].

identified, the most common vaccine serotypes were 6B (17.6%), 19F (11.0%), and 23F (9.3%), and the most common non-vaccine serotypes were 11A (5.1%), 13 (2.7%), and 6C (2.4%) (Fig 1).

## Antibiotics susceptibility of *S. pneumoniae*

All isolates were susceptible to moxifloxacin, levofloxacin, ertapenem, vancomycin, cefotaxime, and cefepime when using the non-meningitis breakpoints (Table 3). We found that 49.7%, 43.9%, 23.3%, 20.8% of isolates were resistant to penicillin (meningitis breakpoints only), tetracycline, trimethoprim/sulfamethoxazole, and chloramphenicol, respectively (Table 3).

Of 1,107 isolates, 61.5% were determined to be NS to ≥1 antibiotic, and by study site, NS was 63.1% (495/784) in Southwest Sumba and 57.6% (186/323) in Gunungkidul (Table 4). Overall NS to ≥1 antibiotic ranged from 56.5–65.4% by age group. In total, 73.9% of the 602 vaccine serotype isolates were NS to ≥1 antibiotic (80.5% in Southwest Sumba and 71.7% in Gunungkidul) (Table 4). By age group, 74.6% of vaccine serotype isolates among children <1 year of age, 76.5% in children 1–2 years of age, and 69.4% in children 3–4 years of age were NS to ≥1 antibiotic. Approximately half of non-vaccine serotype isolates were NS to ≥1 antibiotic; this proportion was higher in Southwest Sumba (52.3%) than in Gunungkidul (37.8%). Among the 445 vaccine serotype isolates NS to ≥1 antibiotic, the most common serotypes were 6B (166/445; 37.3%), 19F (102/445; 22.9%), and 19A (48/445; 10.8%).

MDNS was found in 13.3% of the pneumococcal isolates (147/1,107), and 19.9% of vaccine serotype isolates (120/602). the SXT, tetracyclines, and macrolides were the classes most commonly resistant among MDNS isolates. MDNS in vaccine serotypes varied by study site

**Table 3. Antibiotic susceptibility of pneumococcal Isolates (n = 1,107) obtained from children <5 years of age.**

| Antibiotic Class | Susceptibility Category | | | | | |
| --- | --- | --- | --- | --- | --- | --- |
| | S | | I | | R | |
| | n | % | n | % | n | % |
| **Carbapenems** | | | | | | |
| Ertapenem | 1101 | 99.5 | 6 | 0.5 | 0 | 0.0 |
| Meropenem | 1031 | 93.1 | 47 | 4.2 | 29 | 2.6 |
| **Cephalosporins** | | | | | | |
| Cefepime | | | | | | |
| Non-meningitis breakpoints | 1060 | 95.8 | 47 | 4.2 | 0 | 0.0 |
| Meningitis breakpoints | 1012 | 91.4 | 48 | 4.3 | 47 | 4.2 |
| Cefotaxime | 1080 | 97.6 | 27 | 2.4 | 0 | 0.0 |
| Ceftriaxone | | | | | | |
| Non-meningitis breakpoints | 1051 | 94.9 | 55 | 5.0 | 1 | 0.1 |
| Meningitis breakpoints | 1016 | 91.8 | 35 | 3.2 | 56 | 5.1 |
| Cefuroxime | 1011 | 91.3 | 3 | 0.3 | 93 | 8.4 |
| **Fluoroquinolones** | | | | | | |
| Levofloxacin | 1107 | 100.0 | 0 | 0.0 | 0 | 0.0 |
| Moxifloxacin | 1102 | 99.5 | 5 | 0.5 | 0 | 0.0 |
| **Folate Pathway Antagonists** | | | | | | |
| Trimethoprim/Sulfamethoxazole | 681 | 61.5 | 168 | 15.2 | 258 | 23.3 |
| **Glycopeptides** | | | | | | |
| Vancomycin | 1107 | 100.0 | 0 | 0.0 | 0 | 0.0 |
| **Lincosamides** | | | | | | |
| Clindamycin | 1058 | 95.6 | 1 | 0.1 | 48 | 4.3 |
| **Macrolides** | | | | | | |
| Azithromycin | 1017 | 91.9 | 0 | 0.0 | 90 | 8.1 |
| Erythromycin | 1014 | 91.6 | 3 | 0.3 | 90 | 8.1 |
| **Oxazolidinones** | | | | | | |
| Linezolid[a] | 1105 | 99.8 | 0 | 0.0 | 0 | 0.0 |
| **Penicillins** | | | | | | |
| Amoxicillin/Clavulanic Acid 2:1 | 1065 | 96.2 | 15 | 1.4 | 27 | 2.4 |
| Penicillin | | | | | | |
| Non-meningitis breakpoints | 1056 | 95.4 | 44 | 4.0 | 7 | 0.6 |
| Meningitis breakpoints | 557 | 50.3 | 0 | 0.0 | 550 | 49.7 |
| **Phenicols** | | | | | | |
| Chloramphenicol | 877 | 79.2 | 0 | 0.0 | 230 | 20.8 |
| **Tetracyclines** | | | | | | |
| Tetracycline | 616 | 55.6 | 5 | 0.5 | 486 | 43.9 |

S: susceptible; I: intermediate; R: resistant

[a]Two isolates were considered non-susceptible (NS) (≥4) by the 2022 Clinical and Laboratory Standards Institute (CLSI); CLSI does not provide an interpretive category for intermediate or resistant [19].

(Gunungkidul: 36.4%; Southwest Sumba: 14.3%) and ranged from 19.0% to 21.5% by age group (Table 4). Overall, <5% of non-vaccine serotypes were found to be MDNS. Among the 120 MDNS vaccine serotype isolates, the most common serotypes were 19F (70/120; 58.3%), 6B (22/120; 18.3%), and 19A (13/120; 10.8%). The antibiotic susceptibility of pneumococcal isolates by serotype (overall and by site) are summarized in S2 and S3 Files.

**Table 4. Non-susceptibility and multi-drug non-susceptibility of vaccine and non-vaccine serotype pneumococcal isolates by study site and age group (N = 1107)[a].**

| | NS to one or more antibiotic[b] | | | | | | MDNS[c] | | | | | |
| --- | --- | --- | --- | --- | --- | --- | --- | --- | --- | --- | --- | --- |
| | Vaccine serotype | | Non-vaccine serotype | | Overall[d] | | Vaccine serotype | | Non-vaccine serotype | | Overall[d] | |
| | n/N | % | n/N | % | n/N | % | n/N | % | n/N | % | n/N | % |
| **Total** | 445/602 | (73.9) | 177/376 | (47.1) | 681/1107 | (61.5) | 120/602 | (19.9) | 18/376 | (4.8) | 147/1107 | (13.2) |
| **Study site** | | | | | | | | | | | | |
| Gunungkidul | 124/154 | (80.5) | 51/135 | (37.8) | 186/323 | (57.6) | 56/154 | (36.4) | 4/135 | (3.0) | 63/323 | (19.5) |
| Southwest Sumba | 321/448 | (71.7) | 126/241 | (52.3) | 495/784 | (63.1) | 64/448 | (14.3) | 14/241 | (5.8) | 84/784 | (10.7) |
| **Age group** | | | | | | | | | | | | |
| <1 year | 91/122 | (74.6) | 42/92 | (45.7) | 149/245 | (60.8) | 24/122 | (19.7) | 2/92 | (2.2) | 29/245 | (11.8) |
| 1–2 years | 225/294 | (76.5) | 78/159 | (49.1) | 332/508 | (65.4) | 56/294 | (19.0) | 10/159 | (6.3) | 70/508 | (13.8) |
| 3–4 years | 129/186 | (69.4) | 57/125 | (45.6) | 200/354 | (56.5) | 40/186 | (21.5) | 6/125 | (4.8) | 48/354 | (13.6) |

NS: non-susceptibility; MDNS: multi-drug non-susceptibility

[a]Vaccine serotypes were those included in PCV13 (1, 3, 4, 5, 6A, 6B, 7F, 9V, 14, 18C, 19A, 19F, and 23F). *S. pneumoniae* isolates were deemed to be non-typeable if a serotype could not be determined by cmPCR and Quellung but showed positive results for *lytA* gene. Non-vaccine serotypes were the remaining serotypes (i.e., not included in PCV13 or deemed non-typeable).

[b]Isolates with intermediate or resistant to ≥1 antibiotic were classified as NS.

[c]MDNS was defined as NS to ≥3 classes of antibiotics.

[d]The sum of vaccine and non-vaccine serotypes are not equal to the overall NS or MDNS since non-typeable pneumococcal isolates were not included in the table.

## Discussion

We conducted a nasopharyngeal colonization survey among community dwelling children <5 years of age prior to PCV13 introduction in Indonesia and found 56.3% of the children were colonized with *S. pneumoniae* and 31.8% were colonized with serotypes covered by PCV13. We found a significant difference in the overall (Southwest Sumba = 87.6% vs. Gunungkidul = 30.9%) and PCV13 serotype (Southwest Sumba = 52.6% vs. Gunungkidul = 15.0%) carriage between the peri-urban and rural study sites. These study findings suggest the introduction of PCV13 has the potential to provide large benefits in protecting children against pneumococcal infection and supports the decision made by the Indonesian government to introduce PCV13 into the routine childhood vaccination schedule. PCV13 use in Indonesia started with a demonstration program in 2017 covering two provinces (West Nusa Tenggara and Bangka Belitung) [20], with a broader scale introduction launched in select districts of East and West Java in 2021. Nationwide introduction started in September 2022 [21].

This study was designed as a baseline survey to allow for the evaluation of PCV13 impact on vaccine serotype carriage after widespread introduction of the vaccine. Colonization rates with vaccine serotype strains were higher in Southwest Sumba (52.6% carriage rate; 57.1% of all pneumococcal isolates) than in Gunungkidul (15.0% carriage rates; 47.7% of isolates). The proportion of vaccine serotypes out of pneumococcal strains identified in this study was similar to previous studies from different regions in Indonesia. A cross-sectional study conducted in the Central Lombok Regency in 2012 among healthy children 2–60 months of age found 56% of the pneumococcal strains carried were covered by PCV13 [22]. In a 2016 study conducted in three regions (Bandung, West Java; Central Lombok Regency; Padang, West Sumatra) of Indonesia, 46.3% of the isolates identified from children aged 12–24 months in all three study sites belonged to PCV13 serotypes with regional variation identified (36–58%) [23]. More recently, a study conducted in 2019 among children <5 years of age in South Kalimantan found 46% of the carried pneumococcal strains to be PCV13 serotypes [24].

The most common vaccine serotypes identified in our study (6B, 19F, 23F, 19A, and 14) were also identified in previous surveys in Indonesia. Several carriage studies conducted in different regions of Indonesia between 1997 and 2019 found these serotypes to be commonly carried among children <5 years of age [22, 23, 25, 26] and children <12 years old with HIV infection [27]. Our study identified *S. pneumoniae* serotype 1 (n = 1) and serotype 4 (n = 4), which have not previously been identified from nasopharyngeal samples in children in Indonesia. While *S. pneumoniae* serotype 1 is a common cause of invasive disease in children, it is rarely isolated from the nasopharynx and is not a commonly carried serotype in children [28, 29]. In general, the distribution of most common vaccine serotypes found in this study and previous studies in Indonesia were similar to those found in many countries prior to PCV introduction and suggests PCV13 introduction will have an impact on disease caused by these commonly circulating serotypes in children [10, 30, 31].

We identified serotypes 6C, 11A, 34, 13, 15B, and 15C as the common non-vaccine serotypes in both study sites. Specifically, serotype 6C was the most common non-vaccine serotype identified in Gunungkidul (11.1% of 323 isolates) and serotype 11A was the most common non-vaccine serotype in Southwest Sumba (5.1% of 784 isolates). Evidence of cross-reactivity between serotypes 6C and 6A has been documented and suggests that a cross-protection against disease caused by serotype 6C from a 6A antigen in 10-valent PCV (PCV10) and PCV13 should be expected [32]. Following introduction of PCVs, other countries have reported increases in the circulation of other non-vaccine serotypes. A series of pneumococcal carriage studies conducted in Fiji among children ≤6 years of age and their caregivers before and after PCV10 introduction found serotype replacement was beginning to emerge three years after vaccine introduction among infants and Indigenous children [33]. The impact of PCV13 introduction was also evaluated among children 12–23 months old and infants 5–8 weeks old in the Lao People's Democratic Republic (Lao PDR) in a pre- and post-PCV13 introduction pneumococcal carriage study [34]. Two years after vaccine introduction, there were early signs of serotype replacement with an increasing trend in non-PCV13 serotype carriage, though it was not significant from the baseline study. When evaluating carriage of individual serotypes, there was a significant increase in carriage of serotype 23A in both infants and children, which has been found to increase post-PCV introduction in invasive and non-invasive infections in other settings [35, 36]. Mongolia, compared to Fiji and Lao PDR, also introduced PCV using a 2+1 schedule and found evidence of serotype replacement in carriage one-year post-introduction [37]. In children 12–23 months of age, there was a 1.6-fold increase in non-PCV13 serotype carriage, and more specifically, a significant increase in carriage of serotypes 15A and 23A. In 5–8-week-old infants, there was no change in the non-PCV13 serotype carriage prevalence, though there was a significant increase in carriage of serotypes 15A and 34. Several countries in Europe, the U.S., and Australia have reported an increasing incidence of disease caused by serotypes 8, 9N, 15A, and 23B after PCV13 introduction [38]. Continued monitoring of the circulating serotypes will be needed following PCV13 introduction to determine whether replacement with non-vaccine serotype occurs as has been observed in other countries. New higher valency conjugate vaccines covering 15 and 20 pneumococcal serotypes have been approved in the United States and other countries, and these vaccines are expected to provide additional benefits to prevent disease caused by some of the strains contributing to replacement disease [39–42].

The prevalence of overall *S. pneumoniae* carriage among children <5 years of age in Southwest Sumba in the East region of Indonesia was almost three times higher than in Gunungkidul in the West region of Indonesia (Southwest Sumba = 87.6% vs. Gunungkidul = 30.9%). Previous colonization studies from different regions in Indonesia also demonstrated a wide geographic variability in the overall pneumococcal carriage prevalence. In the East region of

Indonesia, three studies in children <5 years of age conducted in Lombok Island showed the rates of *S. pneumoniae* carriage were 48%, 46%, and 50% in 1997, 2012, and 2016, respectively [15, 18, 22]. Meanwhile in the West region of Indonesia, the prevalence rates of *S. pneumoniae* in children were 35% in Padang, West Sumatera (2016), 43% in Semarang, Central Java (2010), 46% in Jakarta (2012), and 64% in Bandung, West Java (2016) [18–20]. A systematic review of pneumococcal carriage among children in low and lower-middle-income countries found overall carriage rates ranged from 27–91% in pre-PCV studies depending on the country and health status of the study population [43]. The carriage prevalence in Southwest Sumba was much higher than the reported overall carriage prevalence among children <5 years of age found in other countries in the Southeast Asia (38.0–62.8%) and Western Pacific regions (31.4–68.2%) [43], and was closer to the prevalence reported in other parts of the world including in Pakistan (77.2%), India (74.7% for children with clinical pneumonia), and Mozambique (84.5% for children without pneumonia and 80.5% for children with and without HIV) [10, 28, 30, 44]. These geographic differences between the two study sites (Southwest Sumba rural vs. Gunungkidul peri-urban) in overall pneumococcal carriage rates are likely due to socio-demographic factors, such as crowding, exposure to other young children in the household, and exposure to indoor air pollution [45].

The difference in socio-geographical and environmental conditions between Southwest Sumba (rural) and Gunungkidul (peri-urban) likely contributed to the high variability of pneumococcal colonization prevalence. Although we could not collect most data related to these (except for the primary fuel for cooking where nearly all households in Southwest Sumba used wood only compared to <20% in Gunungkidul), we observed Southwest Sumba has poorer access to clean water and health services, which might lead to poorer hygiene compared to those in Gunungkidul. At the same time, housing density was relatively higher in Southwest Sumba, and if air pollution was taken into consideration, those living in Southwest Sumba are more likely to be exposed to indoor air pollution (woodsmoke from cooking indoors) and outdoor pollution (unpaved dirt roads). Exposure to air pollution has been reported to be associated with increased risk of respiratory infections and higher rates and density of pneumococcal colonization [46].

In this study, we found that over 20% of isolates were resistant to tetracycline, trimethoprim/sulfamethoxazole, chloramphenicol, and penicillin when using the meningitis breakpoints. Penicillin has been identified as one of the most commonly prescribed antibiotics to treat respiratory system disorders in children in Indonesia [47]. It was reported that aminopenicillins and tetracyclines accounted for the majority of the prescribed antibiotics among individuals visiting public healthcare facilities in Surabaya and Semarang, Indonesia [48]. Amoxicillin was the most common antibiotic prescribed in community health centers (puskesmas) followed by trimethoprim/sulfamethoxazole, isoniazid, and tetracycline [49]. Prevalence of NS and MDNS was higher among vaccine serotype strains; vaccine serotypes accounted for 73.9% of strains NS to ≥1 antibiotic and 19.9% of MDNS strains. This is consistent with what was observed in countries prior to the introduction of pneumococcal vaccines [10]. When the majority of resistant infections are due to vaccine serotypes, introduction of PCVs has been shown to lead to reductions in antibiotic resistance by reducing the prevalence of the circulating vaccine serotypes [50]. Similar benefits are expected in Indonesia after widespread introduction of PCV13.

There are no laboratory-based surveillance systems to monitor vaccine-preventable pneumococcal disease in Indonesia; however, an estimated 585,770 (uncertainty range [UR] 505,415–696,315) pneumococcal cases and 8,725 (UR 5655–11,038) pneumococcal deaths occurred in children aged <5 years in 2015 [2]. Monitoring the prevalence of serotype-specific carriage in children in two areas of the Indonesian archipelago prior to and after introduction

of PCV13 will allow us to evaluate changes in the prevalence of colonization due to vaccine serotypes with disease potential [28] and to estimate expected direct and indirect impact of the pneumococcal vaccine program [11]. Certain vaccine serotypes, such as serotypes 1 and 4, are rarely carried, therefore, predicting vaccine impact on disease based on the prevalence of carried serotypes will likely lead to an underestimate of the full PCV benefits.

In conclusion, we found that more than 50% of pneumococcal strains colonizing children in Indonesia are covered by PCV13 and the majority of these vaccine type strains were non-susceptible to one or more commonly prescribed antibiotics. Our results suggest that PCV13, introduced into Indonesia's national infant immunization schedule in 2022, has the potential to reduce pneumococcal disease burden. Continued monitoring of the circulating serotypes will be needed to document PCV13 impact on vaccine serotype carriage and to detect the emergence of non-vaccine serotypes and antibiotic resistance.

## Supporting information

**S1 Fig. Distribution of vaccine serotypes, non-vaccine serotypes, and non-typeable pneumococcal isolates[a] by study site[b].**
(PPTX)

**S2 Fig. Distribution of vaccine serotype, non-vaccine serotype, and non-typeable isolates by age group[a,b].**
(PPTX)

**S3 Fig. Carriage prevalence of *S. pneumoniae* and vaccine serotype *S. pneumoniae* by age group and study site[a,b].**
(PPTX)

**S1 File. Interpretive categories and MIC breakpoints for S. pneumoniae isolates[a].**
(PDF)

**S2 File. Antibiotic non-susceptibility of pneumococcal isolates (N = 1,107) obtained from children aged <5 years by serotype and study site.**
(PDF)

**S3 File. Multi-drug non-susceptibility of pneumococcal isolates (N = 1,107) obtained from children <5 years of age by serotype and study site.**
(PDF)

## Acknowledgments

We sincerely thank Donna Angelina Rade, Diyan Yunanto Setyaji and all of the field staff, the Provincial and District Health Offices, *Puskesmas*, *Posyandu* and *kader Posyandu* in Gunung-kidul and Southwest Sumba and Koperasi Jasa Institut Riset Eijkman management, and Centers for Disease Control and Prevention Indonesia for their contribution in the study.

## Author Contributions

**Conceptualization:** Dodi Safari, Wa Ode Dwi Daningrat, Wisiva Tofriska Paramaiswari, Wisnu Tafroji, Amin Soebandrio, Sri Rezeki Hadinegoro, Fabiana C. Pimenta, Maria da Gloria Carvalho, Tamara Pilishvili.

**Data curation:** Wa Ode Dwi Daningrat, Jennifer L. Milucky, Miftahuddin Majid Khoeri, Wisiva Tofriska Paramaiswari, Wisnu Tafroji, Korrie Salsabila, Yayah Winarti.

**Formal analysis:** Wa Ode Dwi Daningrat, Wisnu Tafroji, Korrie Salsabila, Yayah Winarti, Maria da Gloria Carvalho, Tamara Pilishvili.

**Funding acquisition:** Dodi Safari.

**Investigation:** Dodi Safari, Wa Ode Dwi Daningrat, Jennifer L. Milucky, Miftahuddin Majid Khoeri, Wisiva Tofriska Paramaiswari, Wisnu Tafroji, Korrie Salsabila, Yayah Winarti, Amin Soebandrio, Sri Rezeki Hadinegoro, Ari Prayitno, Fabiana C. Pimenta, Maria da Gloria Carvalho, Tamara Pilishvili.

**Methodology:** Wa Ode Dwi Daningrat, Jennifer L. Milucky, Miftahuddin Majid Khoeri, Wisiva Tofriska Paramaiswari, Wisnu Tafroji, Korrie Salsabila, Yayah Winarti.

**Resources:** Miftahuddin Majid Khoeri, Korrie Salsabila.

**Software:** Lana Childs.

**Supervision:** Dodi Safari, Jennifer L. Milucky, Amin Soebandrio, Sri Rezeki Hadinegoro, Ari Prayitno, Lana Childs, Fabiana C. Pimenta, Maria da Gloria Carvalho, Tamara Pilishvili.

**Validation:** Jennifer L. Milucky, Lana Childs, Fabiana C. Pimenta, Maria da Gloria Carvalho, Tamara Pilishvili.

**Visualization:** Wisiva Tofriska Paramaiswari, Korrie Salsabila, Yayah Winarti, Lana Childs.

**Writing – original draft:** Dodi Safari, Wa Ode Dwi Daningrat, Lana Childs, Tamara Pilishvili.

**Writing – review & editing:** Jennifer L. Milucky, Miftahuddin Majid Khoeri, Wisiva Tofriska Paramaiswari, Wisnu Tafroji, Korrie Salsabila, Yayah Winarti, Amin Soebandrio, Sri Rezeki Hadinegoro, Ari Prayitno, Fabiana C. Pimenta, Maria da Gloria Carvalho.

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
