## [Decision Letter · Decision Letter 0]

12 Sep 2023

PONE-D-23-21406Nasopharyngeal carriage of Streptococcus pneumoniae among children <5 years of age in Indonesia prior to pneumococcal conjugate vaccine introductionPLOS ONE

Dear Dr. Safari,

Thank you for submitting your manuscript to PLOS ONE. After careful consideration, we feel that it has merit but does not fully meet PLOS ONE’s publication criteria as it currently stands. Therefore, we invite you to submit a revised version of the manuscript that addresses the points raised during the review process.

The manuscript has been assessed by two reviewers. Their comments are available below. The reviewers have raised a number of concerns about the methodology and the data, they recommend revisions to provide a fuller outline of the methodology and main results.

Please carefully revise the manuscript to address all the points raised by the two reviewers.

The authors declare no competing interests. However, one of the authors indicates as the current address that of a company that manufactures one of the pneumococcal conjugate vaccines. Please clarify.

We look forward to receiving your revised manuscript.

Kind regards,

Jose Melo-Cristino, M.D., Ph.D.

Academic Editor

PLOS ONE

2. This manuscript describes an observational study. Please consider whether this manuscript meets PLOS ONE's guidelines on observational studies involving human subjects https://journals.plos.org/plosone/s/submission-guidelines. If you would like additional assistance evaluating this manuscript, PLOS ONE Staff Editors and Section Editors have developed a tool https://storage.googleapis.com/genweb.plos.org/RR/EditorResources_CSSAssessment.pdf for you to consider as you determine whether the manuscript should be sent for external peer review. If you feel that the quality of the manuscript does not meet the minimum requirements outlined by this tool, please consider rejecting the manuscript before peer review, ensuring that the decision is justified according to PLOS ONE’s publication criteria. Please contact plosone@plos.org with any questions or concerns.

Reviewers' comments:

Reviewer's Responses to Questions

**Comments to the Author**

1. Is the manuscript technically sound, and do the data support the conclusions?

Reviewer #1: Yes

Reviewer #2: Yes

2. Has the statistical analysis been performed appropriately and rigorously? 

Reviewer #1: I Don't Know

Reviewer #2: I Don't Know

3. Have the authors made all data underlying the findings in their manuscript fully available?

Reviewer #1: Yes

Reviewer #2: Yes

4. Is the manuscript presented in an intelligible fashion and written in standard English?

Reviewer #1: Yes

Reviewer #2: Yes

5. Review Comments to the Author

Reviewer #1: The authors of this paper describe a pneumococcal carriage study among children aged <5 years old before the introduction of PCV13, in Indonesia. The authors describe pneumococcal carriage rates, risk factors associated with pneumococcal carriage, serotype distribution as well as antibiotic susceptibility. The authors compare two areas of Indonesia, one rural in the East region and one peri-urban in the West region. This study will contribute to the establishment of a baseline that may be used to monitor how PCV13 impact on colonization in this population.

I have some major comments and minor comments.

Major comments:

1. Please clarify in the Introduction if PCV13 was the first PCV or to be ntroduced in Indonesia. Before 2017, no PCV has been introduced in Indonesia?

2. In the methods, in the study design and population section, a brief description of each region in terms of number of habitants/population density and living conditions may be useful to help put the findings into context.

3. I have some concerns regarding the identification of non-typeable isolates:

- Are they optochin >14 and bile positive? Please clarify.

- Are they non-typeable isolates because a serotype could not be determined or are they non-capsulated isolates?

- There are some strains of the Mitis group lytA positive that are not pneumococcus. Therefore, I would recommend, in addition to lytA real-time PCR, to perform real-time PCR targeting piaB or SP2020 genes. True pneumococci will be positive for both lytA and piaB. Also, a good indication of non-capsulated isolates would be positive results of lytA and SP2020, but negative for piaB (Please see Miellet, 2023, Front Microbiol, doi: 10.3389/fmicb.2023.1122276).

In addition, you could also perform an assay to detect non-capsulated pneumococci based on a PCR targeting lytA, cpsA, aliB-like ORF2, and 16S rDNA genes, plus a restriction fragment length polymorphism assay to differentiate typical from atypical lytA, as described in Simoes et al, 2011, Diagn Microbiol Infect Dis doi: 10.1016/j.diagmicrobio.2011.07.009.

Given the high number of NT isolates found in this study, it may be relevant to clarify these points.

4. Are the differences in carriage prevalence in the two regions significantly different? Given the percentages I would guess that they are significantly different. Please indicate the p-value in the text.

5. Did you perform the analysis of risk factors associated with pneumococcal carriage by geographic location? Please clarify. Could be interesting to do that and see and what kind of results would the authors obtain.

6. I would recommend that the fold-value as well as the p-values for the characteristics that are associated to pneumococcal carriage in the multivariate analysis should be indicated in the main text in the appropriated section.

7. The authors mentioned that children who do not attend day-care centers have higher carriage prevalence. I find it a curious result. On average how many hours per day do children spend in day-cares? Can the authors speculate about this result?

8. Since the comparison between rural and peri-urban regions is one of the main focuses of the study, I encourage the authors to add more discussion on this topic in the discussion.

Minor comments:

1. Could be potentially interesting for some readers to have a table with antimicrobial non-susceptibility associate with each serotype that the authors have found. It could be added in supplementary material

2. Line 118: Replace “during” by “between”.

3. Line 212: Replace “during” by “between”.

4. Line 219: Delete “a”.

Reviewer #2: Article describing the nasopharyngeal carriage S. pneumoniae in Indonesian children under than 5 years old in 2017. The article is well written, methodology is correct and the discussion is supported by the results obtained.

Major comments

1. The main inconvenient of this work is that isolates were collected six years ago, in 2017. Together with the fact that already there are some articles reporting pneumococcal carriage in Indonesia (including Tenggara region and Yogyakarta) previous to PCV13 (references 8, 9, 18, 20, 21 and others) but also post-PCV13 (references 16) makes the study a little old-fashioned. If the authors have data on post-PCV13 it would have been more interesting to observe the supposed changes after vaccine introduction.

2. Line 145. Serotyping. “S. pneumoniae isolates were serotyped by conventional multiplex polymerase chain reaction (cmPCR) based testing [12] followed by Quellung reaction (Staten Institute, Denmark).”

In the reference of Carvalho M da G, et al. only the primers for 13 serotypes/serogroups are described. In fact, they use a “A conventional sequential multiplex PCR able to detect a total of 40 serotypes” and do a reference to the other PCRs used.

Please, use the original references for all PCRs. Also, in the results the number of pneumococci that were not serotyped (non-typeable) should be pointed out. How do the authors know that a concrete non-typeable serotype is less prevalent than other known serotypes (around 10% of the isolates were non-typeable according to Figure 1)?

3. PCVs were not included in Indonesian National Vaccination Schedule until 2022 (line 82). But, were there PCVs available in the private market? Had any children from either region been vaccinated? These data should be included. The % of carriage in Gunungkidul children is very low (30.9%) and the % of vaccine serotypes higher in older children. Could this low % carriage and different serotype distribution have any relation with the use of PCVs in the private market?

Minor comments.

Line 150. Why amoxicillin/clavulanic acid (AMC) and not amoxicillin (AMX) alone? Better to put amoxicillin since AMC could induce misuse of AMC that ha much more seaside effects.

I also suggest using the abbreviation of SXT for trimethoprim/sulfamethoxazole (first time mentioned in line 151) and for “folate pathway antagonists” (line 296).

6. PLOS authors have the option to publish the peer review history of their article (what does this mean?). If published, this will include your full peer review and any attached files.

Reviewer #1: No

Reviewer #2: No

---

## [Author Response · Author response to Decision Letter 0]

2 Nov 2023

Rebuttal Letter

Editor:

The manuscript has been assessed by two reviewers. Their comments are available below. The reviewers have raised a number of concerns about the methodology and the data, they recommend revisions to provide a fuller outline of the methodology and main results.

RESPONSE: We thank both reviewers for their review and comments. We responded to all of their comments and revised the manuscript accordingly.

The authors declare no competing interests. However, one of the authors indicates as the current address that of a company that manufactures one of the pneumococcal conjugate vaccines. Please clarify.

RESPONSE: Tamara Pilishvili was an employee of the US Centers for Disease Control and Prevention during the data collection, data analyses, and the drafting of this manuscript. She recently accepted a job with GSK and as of February 2023 is an employee of GSK. GSK manufactures a 10-valent pneumococcal conjugate vaccine (PCV10, Synflorix), and this vaccine has never been licensed in the US, nor has it been introduced in Indonesia.

RESPONSE: We revised the manuscript according PLOS ONE's style requirements

2. This manuscript describes an observational study. Please consider whether this manuscript meets PLOS ONE's guidelines on observational studies involving human subjects https://journals.plos.org/plosone/s/submission-guidelines. If you would like additional assistance evaluating this manuscript, PLOS ONE Staff Editors and Section Editors have developed a tool https://storage.googleapis.com/genweb.plos.org/RR/EditorResources_CSSAssessment.pdf for you to consider as you determine whether the manuscript should be sent for external peer review. If you feel that the quality of the manuscript does not meet the minimum requirements outlined by this tool, please consider rejecting the manuscript before peer review, ensuring that the decision is justified according to PLOS ONE’s publication criteria. Please contact plosone@plos.org with any questions or concerns.

RESPONSE: This manuscript meets PLOS ONE's guidelines on observational studies involving human subjects 

RESPONSE: We updated the ORCID ID for corresponding author.

RESPONSE: We confirm to include captions for our Supporting Information files at the end of your manuscript, and update any in-text citations to match accordingly.

Reviewer #1: 

Reviewer #1: The authors of this paper describe a pneumococcal carriage study among children aged <5 years old before the introduction of PCV13, in Indonesia. The authors describe pneumococcal carriage rates, risk factors associated with pneumococcal carriage, serotype distribution as well as antibiotic susceptibility. The authors compare two areas of Indonesia, one rural in the East region and one peri-urban in the West region. This study will contribute to the establishment of a baseline that may be used to monitor how PCV13 impact on colonization in this population.

I have some major comments and minor comments.

Major comments:

1. Please clarify in the Introduction if PCV13 was the first PCV or to be introduced in Indonesia. Before 2017, no PCV has been introduced in Indonesia?

RESPONSE: Thank you for your concern. We confirm that PCV13 was the first PCV to be introduced in Indonesia as a demonstration project in Lombok Island in the West Nusa Tenggara province in 2017. The demonstration project only occurred in a limited geographic area (Lombok Island) and the location of this pilot introduction did not overlap with the two study sites included in this study (Gunungkidul in Yogyakarta and Southwest Sumba in East Nusa Tenggara). In 2021, PCV13 introduction was expanded to select districts in East and West Java with nationwide introduction starting in 2022. Gunungkidual and Southwest Sumba were a part of the national PCV13 introduction in 2022. PCVs were only available through a private market and based on the data we collected on PCV vaccination history from enrolled children, we found that only 0.2% (4/1822) had received ≥1 dose of PCV. 

We made the following edits in the introduction:

Lines 79-89 (revised version with track changes): Prior to 2017, 10-valent PCV (PCV10) and the 13-valent PCV (PCV13) were available in Indonesia as part of a private service in hospitals. In 2017, the government of Indonesia introduced PCV13 in a limited geographic area, Lombok Island in the West Nusa Tenggara province, as a demonstration project using a schedule of two primary doses at two and three months of age followed by a booster at 12 months (2+1 schedule) [8]. In 2021, PCV13 was introduced as a part of the national program and launched in select districts in East and West Java and nationwide introduction starting in 2022.

2. In the methods, in the study design and population section, a brief description of each region in terms of number of habitants/population density and living conditions may be useful to help put the findings into context.

RESPONSE: Thank you for this suggestion. We have added the following details to the study design and population section of the methods.

Lines 127-135 (revised version with track changes): Gunungkidul district, Yogyakarta province, has an estimated total population of 770,880 in 2022, with 46,958 (6%) children <5 years of age. Southwest Sumba district in the province of East Nusa Tenggara has an estimated total population of 308,106 in 2022, with 41,334 (13%) children <5 years of age. The district of Southwest Sumba is primarily comprised of rural sub-districts with less access to health services, clean water, and education. Although parts of Gunungkidul district were also categorized as rural by Statistics Indonesia, generally, the district has better access to health services, clean water, and education compared to Southwest Sumba.

3. I have some concerns regarding the identification of non-typeable isolates:

- Are they optochin >14 and bile positive? Please clarify.

RESPONSE: Thank you for your concerns. We confirm that all isolates we defined as non-typeable pneumococcal isolates were indeed confirmed with bile solubility test, diameter of optochin was estimated >14 mm, and positive for lytA gene detection by qPCR test.

4. Are they non-typeable isolates because a serotype could not be determined or are they non-capsulated isolates?

RESPONSE: The non-typeable pneumococcal isolates are negative for quellung test (there is no positive pool antisera, representing all serotypes, detected by quellung). The pneumococcal isolates with negative reaction for quellung antisera, also tested negative for 41 conventional multiplex PCR assays encompassing 70 pneumococcal serotypes (Da Gloria Carvalho et al. 2010. J. Clin. Microbiol. 48: 1611-1618). Also see: Table 1: List of oligonucleotide primers used in 41 conventional multiplex PCR assays for pneumococcal serotype deduction of 70 serotypes (cdc.gov) at: Pneumococcus Streptococcus Lab Resources and Protocols | CDC.

5. There are some strains of the Mitis group lytA positive that are not pneumococcus. Therefore, I would recommend, in addition to lytA real-time PCR, to perform real-time PCR targeting piaB or SP2020 genes. True pneumococci will be positive for both lytA and piaB. Also, a good indication of non-capsulated isolates would be positive results of lytA and SP2020, but negative for piaB (Please see Miellet, 2023, Front Microbiol, doi: 10.3389/fmicb.2023.1122276). 

RESPONSE: Thank you very much for your suggestions. We determined the non-typeable by combination of susceptibility to optochin disc, bile solubility test, lytA detection by qPCR, and quellung antisera negative in consecutive order. We used bile solubility to distinguish non-typeable isolates of S. pneumoniae from non-pneumococcus including S. pseudopneumoniae and S. mitis by considering previous report mentioned S. pseudopneumoniae and S. mitis were insoluble in bile solubility. 

6. In addition, you could also perform an assay to detect non-capsulated pneumococci based on a PCR targeting lytA, cpsA, aliB-like ORF2, and 16S rDNA genes, plus a restriction fragment length polymorphism assay to differentiate typical from atypical lytA, as described in Simoes et al, 2011, Diagn Microbiol Infect Dis doi: 10.1016/j.diagmicrobio.2011.07.009.

Given the high number of NT isolates found in this study, it may be relevant to clarify these points.

RESPONSE: Thank you for your inputs and suggestions. we consider this manuscript to focus on describing the prevalence of S. pneumoniae before pneumococcal vaccine introduction which is isolated from the nasopharynx of children. We will elaborate further about non-typeable in the next manuscript focusing on the non-typeable isolates. 

7. Are the differences in carriage prevalence in the two regions significantly different? Given the percentages I would guess that they are significantly different. Please indicate the p-value in the text.

RESPONSE: Thank you for the suggestion. Yes, the carriage prevalence in the two regions was significantly different. We added the p-value to the results section of the text.

Lines 244-246 (revised version with track changes): We found the carriage prevalence was 87.6% and 30.9% among children in Southwest Sumba (714/815) and Gunungkidul (311/1,007) (P<0.0001), respectively (Table 1).

8. Did you perform the analysis of risk factors associated with pneumococcal carriage by geographic location? Please clarify. Could be interesting to do that and see and what kind of results would the authors obtain.

RESPONSE: Thank you for the question. We did perform the risk factor analysis by site; however, we found there were not sufficient differences in the ORs for the characteristics by site to have the models presented separately. 

9. I would recommend that the fold-value as well as the p-values for the characteristics that are associated to pneumococcal carriage in the multivariate analysis should be indicated in the main text in the appropriated section.

RESPONSE: Thank you for the suggestions. We made the following additions to the text:

Lines 265-277 (revised version with track changes): In a multivariate analysis, the odds of pneumococcal carriage varied significantly by several characteristics: children aged 1–2 years had a 1.7-fold-increased odds compared to children aged <1 year (P=0.0008), households with the presence of other children <5 years old had a 1.9-fold-increased odds compared to those without (P<0.0001), households with 4–6 persons had a 0.8-fold-decreased odds compared to households with 2–3 persons (P=0.012), households using wood only as the primary fuel source had a 4.8-fold-increased odds (P<0.0001) while households using wood with any other source had a 0.8-fold-decreased odds (P<0.0001) compared to households using LPG or kerosene only, exposure to cigarette smoke in the household was associated with a 0.8-fold-decreased odds (P=0.034), and the presence of ≥1 symptom of respiratory illness was associated with a 2.8 fold-increased odds (P<0.0001) (Table 2).

10. The authors mentioned that children who do not attend day-care centers have higher carriage prevalence. I find it a curious result. On average how many hours per day do children spend in day-cares? Can the authors speculate about this result?

RESPONSE: Thank you for the questions. There may be variability in the number of hours children spend in daycare centers across Indonesia; however, generally it is at least 120, 360, and 900 minutes (6 hours) per week for children aged less than 2, 2-4, and 4-6 years old, respectively. Depending on the daycare centers and parents’ flexibility, it’s usually around 2-3 hours per day. We found a higher proportion of children attended daycare in Gunungkidul (30.7%), the peri-urban site, compared to Southwest Sumba, the rural site (17.8%, P<0.0001). The significant difference is likely due to a higher proportion of mothers working outside of the home in the peri-urban site compared to the rural site. When looking at the pneumococcal carriage prevalence by participant characteristics with the sites combined, it is possible the higher carriage prevalence found among children who do not attend daycare is being affected by the higher pneumococcal carriage prevalence in Southwest Sumba, the site with lower daycare attendance. In Table 1, we present the pneumococcal carriage prevalence overall and by study site for the participant characteristics. Pneumococcal carriage among children who do and do not attend daycare does not vary significantly when stratified by study site. In Gunungkidul, pneumococcal carriage prevalence was 32.0% among children who attended daycare compared to 30.4% among those who did not (p=0.598). While the pneumococcal carriage prevalence is higher in Southwest Sumba compared to Gunungkidul, it did not vary among children who attended daycare (88.3%) compared to those who did not (87.5%, p=0.788). Because fewer children attended daycare in Southwest Sumba, the higher pneumococcal carriage prevalence found in this site may be driving the difference when the study sites are combined. While the pneumococcal carriage prevalence among children who did not attend daycare was higher with the study site combined, it should be considered in the context of the variability of this characteristic and carriage prevalence by study site.

We added the following text to the results to make note that there is no longer a difference when stratified by site:

Lines 252-256 (revised version with track changes): Children who did not attend daycare also had a slightly higher carriage prevalence than those who did (58.3%; 798/1,368 vs. 50.0%; 227/454); however, when stratified by site there was no difference in the carriage prevalence by daycare attendance (Gunungkidul: 32.0%; 99/309 vs. 30.4%; 212/698; Southwest Sumba: 88.3%; 128/145 vs. 87.5%; 586/670).

11. Since the comparison between rural and peri-urban regions is one of the main focuses of the study, I encourage the authors to add more discussion on this topic in the discussion.

RESPONSE: Thank you for your suggestions. We have made the following addition to the discussion:

Lines 448-465: The difference in socio-geographical and environmental conditions between Southwest Sumba (rural) and Gunungkidul (peri-urban) likely contributed to the high variability of pneumococcal colonization prevalence. Although we could not collect most data related to these (except for the primary fuel for cooking where nearly all households in Southwest Sumba used wood only compared to <20% in Gunungkidul), we observed Southwest Sumba has poorer access to clean water and health services, which might lead to poorer hygiene compared to those in Gunungkidul. At the same time, housing density was relatively higher in Southwest Sumba, and if air pollution was taken into consideration, those living in Southwest Sumba are more likely to be exposed to indoor air pollution (woodsmoke from cooking indoors) and outdoor pollution (unpaved dirt roads). Exposure to air pollution has been reported to be associated with increased risk of respiratory infections and higher rates and density of pneumococcal colonization [44]. 

Minor comments:

1. Could be potentially interesting for some readers to have a table with antimicrobial non-susceptibility associate with each serotype that the authors have found. It could be added in supplementary material

RESPONSE: Thank you for this suggestion. We propose the addition of two additional tables in the supplement for the editor’s consideration. We made mention of these additional tables in the below lines of the results:

Lines 328-330 (revised version with track changes): The antibiotic susceptibility of pneumococcal isolates by serotype (overall and by site) are summarized in S2 and S3 Tables. 

2. Line 118: Replace “during” by “between”.

RESPONSE: Thank you very much for your input. We have revised the manuscript according to your suggestion.

3. Line 212: Replace “during” by “between”.

RESPONSE: Thank you very much for your input. We have revised the manuscript as suggested.

4. Line 219: Delete “a”.

RESPONSE: Thank you very much for the input; we have deleted it according to your suggestion.

Reviewer #2: 

Reviewer #2: Article describing the nasopharyngeal carriage S. pneumoniae in Indonesian children under than 5 years old in 2017. The article is well written, methodology is correct and the discussion is supported by the results obtained.

Major comments

1. The main inconvenient of this work is that isolates were collected six years ago, in 2017. Together with the fact that already there are some articles reporting pneumococcal carriage in Indonesia (including Tenggara region and Yogyakarta) previous to PCV13 (references 8, 9, 18, 20, 21 and others) but also post-PCV13 (references 16) makes the study a little old-fashioned. If the authors have data on post-PCV13 it would have been more interesting to observe the supposed changes after vaccine introduction.

RESPONSE: Thank you for this comment. Despite the publications mentioned by the reviewer, data on pneumococcal disease burden in Indonesia are limited, and there are no laboratory-based surveillance systems capable of serving as a platform for PCV impact evaluations. None of the articles mentioned by the reviewer have shown such large variability in the pneumococcal carriage prevalence across Indonesia pre-PCV13 introduction as we found in this study. The article mentioned by the reviewer post-PCV13 focused only on the site of the PCV13 demonstration program, and it was conducted prior to national introduction of PCV13. Since national introduction did not occur until 2022, we are not aware of any published studies evaluating the impact of nationwide PCV13 introduction. While the study was conducted in 2017, the data are still relevant to establish a baseline pneumococcal carriage prevalence pre-national PCV13 introduction given the limited data available in Indonesia. 

2. Line 145. Serotyping. “S. pneumoniae isolates were serotyped by conventional multiplex polymerase chain reaction (cmPCR) based testing [12] followed by Quellung reaction (Staten Institute, Denmark).”

In the reference of Carvalho M da G, et al. only the primers for 13 new serotypes/serogroups are described at the time of publication. In fact, they use a “A conventional sequential multiplex PCR able to detect a total of 40 serotypes” and do a reference to the other PCRs used seeat www.cdc.gov/ncidod/biotech/strep/pcr.htm it referenced to a table with 41 conventional multiplex PCR assays encompassing 70 pneumococcal serotypes. Pneumococcus Streptococcus Lab Resources and Protocols | CDC

Please, use the original references for all PCRs. Also, in the results the number of pneumococci that were not serotyped (non-typeable) should be pointed out. How do the authors know that a concrete non-typeable serotype is less prevalent than other known serotypes (around 10% of the isolates were non-typeable according to Figure 1)?

RESPONSE: Thank you very much for your revision and suggestions. We have revised the reference and have included the original reference as reviewer addressed. You can also find table with assays and all references of each StrepLab home page:

 Table 1: List of oligonucleotide primers used in 41 conventional multiplex PCR assays for pneumococcal serotype deduction of 70 serotypes (cdc.gov)

3. PCVs were not included in Indonesian National Vaccination Schedule until 2022 (line 82). But, were there PCVs available in the private market? Had any children from either region been vaccinated? These data should be included. The % of carriage in Gunungkidul children is very low (30.9%) and the % of vaccine serotypes higher in older children. Could this low % carriage and different serotype distribution have any relation with the use of PCVs in the private market?

RESPONSE: Thank you for these comments and questions. Regarding the availability of PCVs on the private market, we made the following addition to the text: 

Lines 79-89 (revised version with track changes): Prior to 2017, 10-valent PCV (PCV10) and the 13-valent PCV (PCV13) were available in Indonesia as part of a private service in hospitals. In 2017, the government of Indonesia introduced PCV13 in a limited geographic area, Lombok Island in the West Nusa Tenggara province, as a demonstration project using a schedule of two primary doses at two and three months of age followed by a booster at 12 months (2+1 schedule) [8]. In 2021, PCV13 was introduced as a part of the national program and launched in select districts in East and West Java and nationwide introduction starting in 2022.

Minor comments.

Line 150. Why amoxicillin/clavulanic acid (AMC) and not amoxicillin (AMX) alone? Better to put amoxicillin since AMC could induce misuse of AMC that ha much more seaside effects.

RESPONSE: We used the Sensititre plate STP6F for determining the antimicrobial susceptibility profile of our isolates. In this commercial panel, the amoxicillin is provided in combination with clavulanate (Amoxicillin/clavulanic acid) and the amoxicillin alone is not provided in the panel. We reported the antibiotics covered by the panel. 

I also suggest using the abbreviation of SXT for trimethoprim/sulfamethoxazole (first time mentioned in line 151) and for “folate pathway antagonists” (line 296).

RESPONSE: Thank you for the suggestion, we have revised as suggested in line 166 and line 323.

---

## [Decision Letter · Decision Letter 1]

21 Nov 2023

PONE-D-23-21406R1Nasopharyngeal carriage of Streptococcus pneumoniae among children <5 years of age in Indonesia prior to pneumococcal conjugate vaccine introductionPLOS ONE

Dear Dr. Safari,

Thank you for submitting your manuscript to PLOS ONE. After careful consideration, we feel that it has merit but does not fully meet PLOS ONE’s publication criteria as it currently stands. Therefore, we invite you to submit a revised version of the manuscript that addresses the points raised during the review process.

Please answer the two points raised by reviewer #2.

We look forward to receiving your revised manuscript.

Kind regards,

Jose Melo-Cristino, M.D., Ph.D.

Academic Editor

PLOS ONE

Journal Requirements:

Reviewers' comments:

Reviewer's Responses to Questions

**Comments to the Author**

1. If the authors have adequately addressed your comments raised in a previous round of review and you feel that this manuscript is now acceptable for publication, you may indicate that here to bypass the “Comments to the Author” section, enter your conflict of interest statement in the “Confidential to Editor” section, and submit your "Accept" recommendation.

Reviewer #1: All comments have been addressed

Reviewer #2: (No Response)

2. Is the manuscript technically sound, and do the data support the conclusions?

Reviewer #1: Yes

Reviewer #2: Partly

3. Has the statistical analysis been performed appropriately and rigorously? 

Reviewer #1: I Don't Know

Reviewer #2: Yes

4. Have the authors made all data underlying the findings in their manuscript fully available?

Reviewer #1: Yes

Reviewer #2: Yes

5. Is the manuscript presented in an intelligible fashion and written in standard English?

Reviewer #1: Yes

Reviewer #2: Yes

6. Review Comments to the Author

Reviewer #1: The authors have addressed all comments raised by the reviewer who is satisfied with the answers provided.

The paper meets the PLOS ONE criteria for publication and is now acceptable for publication.

Reviewer #2: Question 2 of R1.

Authors have not included the original references used for PCR-serotyping described in the CDC web page. They only include one of the original references, but others as Pai et al . 2006, J. Clin. Microbiol. 44: 124-131 or Pimenta et al . 2009. J. Clin. Microbiol. I7: 2353-2354, Menezes et al. 2013, J. Clin. Microbiol. 51(7):2470-1 have not been included.

Besides, the next question has not been answered:

How do the authors know that a concrete non-typeable serotype is less prevalent than other known serotypes (around 10% of the isolates were non-typeable according to Figure 1)?

7. PLOS authors have the option to publish the peer review history of their article (what does this mean?). If published, this will include your full peer review and any attached files.

Reviewer #1: No

Reviewer #2: No

---

## [Author Response · Author response to Decision Letter 1]

21 Dec 2023

Rebuttal Letter

Reviewer #2: Question 2 of R1.

Authors have not included the original references used for PCR-serotyping described in the CDC web page. They only include one of the original references, but others as Pai et al . 2006, J. Clin. Microbiol. 44: 124-131 or Pimenta et al . 2009. J. Clin. Microbiol. I7: 2353-2354, Menezes et al. 2013, J. Clin. Microbiol. 51(7):2470-1 have not been included.

RESPONSE: Thank you very much for your revision and suggestions. We have revised the reference and have included the original reference as reviewer addressed. We have added these references to line 152 to 154.

Besides, the next question has not been answered:

How do the authors know that a concrete non-typeable serotype is less prevalent than other known serotypes (around 10% of the isolates were non-typeable according to Figure 1)?

RESPONSE: Thank you for this comment. Regarding your question related to the prevalence of vaccine serotypes, non-vaccine serotypes and non-typeable including how we determine the non-typeable, we would confirm that the non-typeable isolates were determined by following criteria: bile soluble, negative antisera reaction, and positive lytA gene detection as we provided in comment #3 of reviewer #1. These 3 criteria will define the non-typeable Streptococcus pneumoniae (you might call these isolates as concrete non-typeable). The presentation we provided in this manuscript are divided into 3 groups as commonly reported in many publications related to prevalence of pneumococcal serotypes; vaccine serotype prevalence, non-vaccine serotype prevalence, and non-typeable prevalence. We have provided the prevalence of non-typeable in line 288 – 290 “Overall, 11.7% of the pneumococcal isolates were non-typeable; 12.1% (95/784) in Southwest Sumba and 10.5% (34/323) in Gunungkidul (Fig. 1)” while the other 2 groups; vaccine serotypes and non-vaccine serotypes are provided in line 280 – 288 ” Among the 1,107 pneumococcal isolates identified, the most common vaccine serotypes were 6B (17.3%), 19F (12.4%), and 23F (7.8%), and the most common non-vaccine serotypes were 6C (5.0%), 11A (4.0%), and 34 (3.1%). In Gunungkidul, among the 323 pneumococcal isolates identified, the most common vaccine serotypes were 6B (16.4%), 19F (15.8%), and 3 (4.6%), and the most common non-vaccine serotypes were 6C (11.1%), 34 (6.8%), and 15C (3.4%) (Fig 1). In Southwest Sumba, among the 784 pneumococcal isolates identified, the most common vaccine serotypes were 6B (17.6%), 19F (11.0%), and 23F (9.3%), and the most common non-vaccine serotypes were 11A (5.1%), 13 (2.7%), and 6C (2.4%) (Fig 1)”. 

It is common to report the non-typeable separately from the non-vaccine serotypes in many previous publication since the non-typeable is different with non-vaccine serotypes whose capsule and capsular polysaccharide as surface antigen and can react with antisera during quellung serotyping. This capsule and capsular polysaccharide among non-vaccine serotypes is important for surveillance records since the ability of being invasive due to presence of capsule and potential vaccine formulation by identifying the capsular polysaccharide. Different with non-typeable, this group has no capsule and capsular polysaccharide that makes this group do not react with antisera. The non-typeable will expose the surface antigen (due to lack of capsule) to the environment that will be recognized by immune system which makes this group less concern in the surveillance. Therefore, in many publication, the non-typeable prevalence will be separated from the non-vaccine serotypes which are sometimes reported less than the non-typeable group.

---

## [Editor Report · Decision Letter 2]

28 Dec 2023

Nasopharyngeal carriage of Streptococcus pneumoniae among children <5 years of age in Indonesia prior to pneumococcal conjugate vaccine introduction

PONE-D-23-21406R2

Dear Dr. Safari,

We’re pleased to inform you that your manuscript has been judged scientifically suitable for publication and will be formally accepted for publication once it meets all outstanding technical requirements.

Kind regards,

Jose Melo-Cristino, M.D., Ph.D.

Academic Editor

PLOS ONE
---

## [Editor Report · Acceptance letter]

2 Jan 2024

PONE-D-23-21406R2 

PLOS ONE

Dear Dr. Safari, 

I'm pleased to inform you that your manuscript has been deemed suitable for publication in PLOS ONE. Congratulations! Your manuscript is now being handed over to our production team.

Kind regards, 

on behalf of

Prof. Jose Melo-Cristino 

Academic Editor

PLOS ONE